# Does Drinking Coffee and Tea Affect Bone Metabolism in Patients with Inflammatory Bowel Diseases?

**DOI:** 10.3390/nu13010216

**Published:** 2021-01-13

**Authors:** Alicja Ewa Ratajczak, Aleksandra Szymczak-Tomczak, Agnieszka Zawada, Anna Maria Rychter, Agnieszka Dobrowolska, Iwona Krela-Kaźmierczak

**Affiliations:** Department of Gastroenterology, Dietetics and Internal Diseases, Poznan University of Medical Sciences, 61-701 Poznań, Poland; aleksandra.szymczak@o2.pl (A.S.-T.); a.zawada@ump.edu.pl (A.Z.); a.m.rychter@gmail.com (A.M.R.); agdob@ump.edu.pl (A.D.)

**Keywords:** Crohn’s disease, colitis, ulcerative, caffeine, *Camellia sinensis*

## Abstract

Patients suffering from Crohn’s disease and ulcerative colitis are at higher risk of osteoporosis due to lower bone mineral density. Risk factors of osteoporosis are divided into unmodifiable, namely, age, gender, genetic factors, as well as modifiable, including diet, level of physical activity, and the use of stimulants. Coffee and tea contain numerous compounds affecting bone metabolism. Certain substances such as antioxidants may protect bones; other substances may increase bone resorption. Nevertheless, the influence of coffee and tea on the development and course of inflammatory bowel diseases is contradictory.

## 1. Introduction

The intake of products with caffeine may affect bone metabolism [1], whereas excessive consumption of coffee and tea constitutes a modifiable risk factor of osteoporosis. Therefore, a change of detrimental habits may decrease the risk of osteoporosis development [2,3,4].

The incidence of inflammatory bowel diseases (IBD)—Crohn’s disease (CD) and ulcerative colitis (UC)—has increased in the recent decades. The highest IBD morbidity is observed in North America where 20.2/100 inhabitants suffer from CD and 19.2/100,000 suffer from UC, while IBD affects about 2 million people in Europe, and 1.5 million in America. It is vital to notice that UC and CD are generally diagnosed in highly developed countries. The main symptoms of IBD are gastrointestinal complaints, although the disease may affect other systems as well [5,6]. In the Western countries, women suffer from IBD (particularly from Crohn’s disease) more often than men. The etiology of IBD is not fully understood. Although there are several loci associated with CD or UC, which confirms the genetic basis of the disease, the environmental factors may additionally participate in the development of the disease [7]. CD generally starts in the terminal part of the ileum, but may affect any part of the gastrointestinal tract, and the inflammatory elements are non-continuous. On the other hand, UC affects the colon progressing from the distal to the proximal part, and the inflammation is continuous [8].

Even on the basis of the clinical data, the association between drinking coffee and tea and IBD remains unclear. The compounds of the abovementioned beverages such as caffeine may affect IBD patients both positively and negatively. Additionally, coffee increases motor activity of the intestines within 4 min of consumption, and the impact of both beverages on the colon is similar to the consumption of a 1000 kcal meal [1]. Bearing in mind that higher motor activity of the intestines may exacerbate diarrhea in IBD, Rao et al. reported that caffeinated coffee increased motor activity of the colon similarly to consumption of a meal, 60% and 23% more than water and decaffeinated coffee, respectively [2]. 

Moreover, caffeine may inhibit appetite, thus increasing the risk of malnutrition among patients suffering from IBD. It is vital to notice that coffee may decrease the tension of the lower esophageal sphincter leading to the exacerbation of the gastroesophageal reflux disease. Additionally, coffee may be harmful for CD patients with inflammation in the upper gastrointestinal tract [1], as it may increase insomnia and elevate the level of stress hormones.

Osteoporosis is a chronic bone disease with low bone mineral density (BMD) which may cause fragility fractures, disability and decrease the quality of life. One third and one fifth of women and men, respectively, aged over 50 years, suffer an osteoporotic fracture [9,10,11]. The ageing of the population causes an increase in the number of people who suffer from osteoporosis—it is estimated that nowadays 200 million people report this disorder [12]. The prevalence of osteoporosis varies in different regions of the world [13]. The risk factors of osteoporosis include female gender, age, BMI (body mass index), low physical activity, inadequate diet (insufficient calcium and vitamin D intake), occurrence of a fracture in the past, low muscle mass, genetic factors, diagnosis of osteoporosis in the immediate family, administration of certain drugs (such as steroids), and occurrence of certain diseases, including IBD [1]. Postmenopausal women have a higher risk of osteoporosis and many studies on osteoporosis concern this group [3,12,14,15].

Moreover, peak bone mass is independent of other genetic factors, as well as of nutrition, race, region of life, and environmental factors. Additionally, risk factors for IBD patients comprise gut-bone immune signaling and pathogenic microbiota [16,17]. 

IBD patients are at an increased risk of low BMD and bone fracture [18]. In fact, osteoporosis or osteopenia affects about 18–42% of adults and 20–50% of children suffering from IBD. According to a Polish study by Krela-Kaźmierczak et al., osteoporosis of the femoral neck and lumbar spine occurs in 45.3% of women and 24.5% of men.

## 2. Caffeine and Tea—Bone Metabolism, Calcium, and Phosphate Management

The impact of coffee consumption on bone metabolism remains controversial. The caffeine contained in coffee may affect BMD by means of numerous mechanisms, as it increases urinary calcium excretion, inhibits proliferation of osteoblasts and bone healing process leading to an elevated risk of fractures [19,20,21].

Cytotoxicity of caffeine can be caused by inducing apoptosis [22], since caffeine stimulates the formation of reactive oxygen spices, hence inducing the apoptosis cascade. As a result, cysteine proteases (caspases) and members of the BCL-2 family are activated. Caspases and members of the BCL-2 family regulate the change of mitochondrial membrane permeability and release cytochrome C due to modulating the permeability of the outer mitochondrial membrane. Additionally, caffeine may inhibit the anti-apoptosis pathway of osteoblasts which involves ERK (extracellular signal-regulated kinases) and Akt (protein kinase B, PKB) [23].

The animal study demonstrated that caffeine decreased the formation of mineralized nodules and osteoblast colonies in Wistar rats. Additionally, the activity of LDH (lactate dehydrogenase) and PGE2 (prostaglandin E2), which is produced by osteoblasts, was also reduced. In fact, this research suggests a negative impact of caffeine on metabolism and vitality of bone cells [24]. Caffeine contained in coffee affects osteogenesis through reducing the differentiation of mesenchymal stem cells (MSC) in osteogenic lineages and inhibition of specific gene expression. Differentiation of MSC is controlled by Cbfa1/Runx2 (runt-related transcription factor 2 (RUNX2), also known as core-binding factor subunit alpha-1), which may be regulated by cAMP (cyclic adenosine monophosphate). Therefore, caffeine may increase intracellular cAMP by inhibition of the activity of cAMP phosphodiesterase, which leads to a decrease in cAMP degradation [25]. It is assumed that caffeine takes part in the regulation of expression of the Cbfa1/Runx2 gene and decreases differentiation rate of MSC in osteoblasts.

Caffeine increases urinary calcium excretion and decreases the resorption of calcium in the intestines [26]. On the other hand, it also increases urinary excretion of magnesium, sodium, and chloride, and this process persists for at least 3 h after consumption. Although the hypercalciuric effect is dependent on the caffeine dose, it may be inhibited by the adenosine agonist [27]. This, in turn, may result in a decrease of BMD, particularly in patients who cannot compensate for calcium loss in urine by means of a diet, as well as in patients suffering from IBD due to the change in the mucous lining of the intestines which further leads to decreased intestinal absorption.

Researchers also investigate the impact of coffee drinking on calcium-phosphate balance, which may lead to bone disorders, including osteoporosis [28]. Compounds of coffee, especially caffeine, impair calcium absorption and stimulate calcium excretion [29]. In fact, caffeine increases the excretion of calcium, magnesium, sodium, and chlorides for a minimum of 3 h following consumption [27]. Urinary calcium loss due to caffeine consumption is probably a consequence of decreased renal absorption. Additionally, caffeine effect is proportional to the dose for free fat mass [30]. As Massey et al. report, caffeine intake decreases serum inositol level, which participates in calcium metabolism and may slightly increase calcium excretion and decrease absorption [27]. Furthermore, calcium-phosphate imbalance may decrease BMD. Nevertheless, individuals consuming a recommended daily dose of calcium are not at risk of caffeine impact on bone calcium metabolism [31]. Additionally, theophylline, 1,3-dimethylxanthine, also increases calcium excretion in animals [32]. Table 1 presents some coffee compounds which may affect bone.

## 3. Coffee

### 3.1. Coffee Consumption

Coffee is the most popular non-alcoholic beverage in the world. Most studies regarding the impact of coffee on the risk of development of certain diseases are observational. In fact, data interpretation may be obstructed by other anti-health behaviors associated with coffee consumption such as cigarette smoking and low physical activity [35]. Coffee contains more than one thousand various chemical compounds, including carbohydrates, fats, nitrogen compounds, vitamins, minerals, as well as alkaloids, which are potentially beneficial for health [36]. In fact, caffeine constitutes one of the main and best-known substances in coffee [35]. The content of caffeine in coffee may vary and depends on the method of preparation of the drink, e.g., espresso contains 30–50 mg of caffeine, instant coffee—about 60–85 mg, whereas dripped coffee has between 85–120 mg [37]. However, caffeine is also present in tea leaves, cocoa beans (chocolate), yerba mate leaves, kola nuts, or guarana [38,39,40]. Nowadays, energy shots and carbonated soft beverages, which also contain caffeine and are often consumed by young people, including children, are becoming increasingly popular [41]. Additionally, caffeine may be added to medications such as analgesics [42].

### 3.2. Coffee Consumption and Risk of IBD

The data concerning the impact of coffee on the development of IBD are contradictory. According to Hansen et al., the consumption of three or more cups of coffee did not alter the risk of disease development in comparison to subjects who consumed smaller amounts of coffee [43]. In Asian and Australian populations, coffee intake decreased the risk of UC, although it did not change the risk of CD [44]. Furthermore, in spite of the fact that the meta-analysis showed that coffee consumption might protect individuals from UC and CD, the impact was not significant [45,46]. In fact, more than 70% of patients suffering from IBD declared regular coffee consumption, with 6.5% choosing the decaffeinated variety [47].

Moreover, almost two thirds of the subjects avoiding coffee reported malaise and worsening gastrointestinal symptoms following coffee consumption [47]. No differences were observed in coffee consumption between men suffering from IBD and healthy individuals [48], although according to questionnaire surveys by Gacek et al., more than 67% of IBD patients declared avoiding consuming excessive amounts of coffee and tea [49]. 

### 3.3. Coffee Consumption and the Risk of Osteoporosis in IBD Patients

An animal study demonstrated that a medium and high dose of caffeine decreased the level of serum alkaline and acid phosphatases in rats with ovariectomy-induced osteoporosis [50].

Chau et al. reported that coffee consumption was correlated positively with BMD of the lumbar spine and femoral neck. Moreover, metabolites related to coffee consumption were correlated with bone mineral density [51]. Intake of more than 1000 mL coffee per day increased calcium excretion by 1.6 mmol, whereas consumption of 1–2 cups of coffee every day slightly affected calcium balance [52]. On the other hand, high and moderate intake of coffee correlated with a higher T-score. Researchers observed a trend in a group with moderate consumption, which may suggest the T-score increases as the consumption of coffee increases [53]. Additionally, there was no association between coffee intake and occurrence of fractures or femoral neck fractures in women, although intake of four and more cups of coffee per day decreased BMD by 2–4% when compared to the subjects consuming less than one cup of coffee per day [54]. Moreover, in postmenopausal women, higher consumption of coffee was associated with lower BMD, whereas changes of BMD did not occur in women who consumed one cup of milk every day for most of their lives [55]. The study indicated that moderate consumption of coffee might protect bone loss in postmenopausal women [56], and the risk of osteoporosis development was lower in men (age: 64.85 ± 9.41 years) who consumed moderate amounts of coffee than in the subjects who did not drink this beverage [57]. Al-Othman et al. reported that the serum level of 25(OH)D was not different among groups with various coffee intake levels [58]. Therefore, moderate coffee intake might decrease the risk of osteoporosis in patients with IBD.

## 4. Tea

### 4.1. Tea Consumption

Tea (*Camellia sinensis*) is a plant used worldwide to prepare the beverage. Green tea is particularly appreciated, as it contains numerous antioxidants substances. In its leaves, we can find such substances as catechins, e.g., epigallocatechin 3-gallate, epicatechin, epicatechin 3-gallate, epigallocatechin, theobromine, theophylline, phenolic acid, or caffeine. In fact, flavonoids may account for 30% of the dry matter [59] and owing to their antioxidant properties, tea has been known as the product which protects against diseases and certain tumors [60]. It is vital to notice that green tea is consumed mainly in Asian and Northern African countries and black tea is the most popular in the United States, England, and other Western countries. Moreover, oolong tea is consumed primarily in Taiwan, southern China, and most Eastern countries [61].

### 4.2. Tea Consumption and Risk of IBD

It is generally accepted that consumption of tea protects from the development of UC and CD [45,46,62]. In their study, Du et al. indicate that epigallocatechin 3-gallate decreased cellular and molecular inflammation and intestinal permeability in the inflammation of the intestines [63]. Moreover, inflammatory mediators were reduced in bowels of the mice fed epigallocatechin 3-gallate and suffering from induced inflammation. However, digestion of proteins and fats decreased in the study group, which is unfavorable in patients with IBD [64]. Polyphenols of green tea may decrease inflammation by regulating production of IKK (I kappa B kinase complex), TNFγ (tumor necrosis factor γ), Cox-2 (cyclooxygenase-2), Bcl-2 (B-cell lymphoma 2) and NF-kB (nuclear factor kappa B) [65,66].

Table 2 demonstrates compounds which influence the bone.

### 4.3. Tea Consumption and Risk of Osteoporosis in IBD Patients

Supplementation of 500 mg green tea polyphenols (GTP) increased bone-specific alkaline phosphatase which is a bone formation marker. Additionally, supplementation did not alter the serum calcium level and calcium excretion [69]. In fact, a meta-analysis showed that tea consumption decreased the risk of osteoporosis [70]. According to Zhang et al., subjects drinking tea presented a higher BMD of the hip and femoral neck than non-drinkers. However, there was no association between tea consumption and the total BMD [60]. Furthermore, a Chinese study demonstrated that moderate intake of tea positively affected bone health in women. Nevertheless, higher consumption neither decreased nor increased BMD, and in men, no association between tea intake and BMD was found [71]. Individuals drinking tea had a higher BMD by about 1.9% [72]. Guo et al. investigated whether tea consumption increased BMD; however, the association between the intake of tea and osteoporotic fractures needs more research [73].

## 5. Coffee and Tea Consumption and Microbiota in IBD Patients

On the basis of various studies, intestinal dysbiosis is one of the most vital factors of the IBD pathogenesis [74] and a modification of microbiota is compared to pharmacological treatment. According to Kruis et al., the use of *Escherichia coli* strain Nissle 1917 was equivalent to mesalazine treatment with regard to the relapse [75]. Interestingly, the research topic of the aforementioned research was focused on the influence of nutrients and stimulants on the microbiota and the course of IBD. Moreover, in their study, Ng et al. observed the impact of coffee and tea on the risk of IBD development where multivariate logistic regression indicated that tea consumption was associated with a decreased risk of CD. Additionally, among the Asian population, the intake of coffee and tea was linked with a lower risk of UC [44]. Although the mechanism of the association remains unclear, it is possible that one of the factors may be the impact of coffee and tea on gut microbiota. In fact, coffee and its components (caffeine and chlorogenic acid among others) may affect composition of the microbiota. As Nishitsuji et al. observed, the use of coffee and chlorogenic acid restored balance of short-chain fatty acids (SCFAs) in obese mouse suffering from diabetes and, in consequence, from microbiota dysbiosis [76]. It is worth remembering that SCFAs play a vital role in the regulation of metabolic processes and affect both the immune system and the proliferation of many cells [77]; specifically, butyrate is particularly crucial for colon cells [78]. Furthermore, in their study, Nishitsuji et al. found that the percentage of six microbial genera in gut microbiota was altered [76]. In fact, consumption of coffee affected the silva microbiota by increasing the number of *Streptococcus mitis* and *Streptococcus infantis* [79]. However, there are limited data regarding the negative impact of caffeine on the gut microbiota. According to the study by Kleber Silveira et al., guarana reportedly improved the redox parameter, although caffeine contained in the guarana affected gut microbiota negatively [80].

The pathogenesis of inflammatory bowel diseases is associated with the infiltration of epithelial and submucosal cells. Chitinase-3-like protein 1 (CHI3L1) is a host protein facilitating the connection of bacteria to epithelial cells. Lee et al. observed that blocking CHI3L1 by caffeine, which is an inhibitor of pan-chitinase, may decrease the risk of colitis. In fact, caffeine treatment decreased CHI3L1 mRNA expression which resulted in a reduction of bacterial invasion to walls of the intestines depending on the caffeine dose. The caffeine dosing triggered a weaker response on dextran sulphate sodium (which stimulates colitis), lower body mass loss, better clinical and histological results in mice. Additionally, bacterial translocation to other organs and pro-inflammation cytokines were lower. On the basis of the study by Lee et al., caffeine inhibits acute colitis by the change of bacterial interaction and a decrease in the expression of CHI3L1 [81].

Pu-erh constitutes a good source of polyphenols and caffeine, as well as affects body composition and energy efficiency of various cells. Pu-erh intake decreases inflammation markers in mice on a high-fat diet. Additionally, pu-erh changes the gut bacterial profile, as following the consumption of pu-erh, an increase was observed in *Akkermansia muciniphila* (*A. muciniphila*), increasing lipid oxidation and glucose metabolism, as well as in *Faecalibacterium prausnitzii*, decreasing inflammation response of the liver and gut caused by a high-fat diet [82].

Ginseng tea modified gut microbiota among rats with experimental colitis by means of increasing colonization of *Lactobacillus* and *Bifidobacterium* as well as by inhibiting growth of *Escherichia coli* (*E.coli*). It is vital to notice that catechins of tea also affect chronic bowel inflammation. Only a small amount of catechins is absorbed in the small intestine, and a significant portion is transported to the large intestine where they are broken down and absorbed [83]. Interestingly, catechins may modulate the gut microbiota composition, which plays a role in the regulation of production of metabolites and their biological activity. An in vitro study demonstrated that polyphenols of black tea inhibit growth of the pathogens harmful for patients suffering from IBD and with suppressed immune systems. In fact, polyphenols control the growth of *Helicobacter pylori*, *Staphylococcus aureus*, *Escherichia coli* O157:H7, *Salmonella typhimurium* DT104, *Pseudomonas aeruginosa* [84].

A. muciniphila is a bacterium which affects the production of SCFA as well as the stimulation of mucus production by goblet cells improving the integrity of the intestinal mucosal barrier. Additionally, *A. muciniphila* decreases the number of Firmicutes and Clostridia, promoting bowel homeostasis [85]. It is vital to notice that polyphenols in tea and other products may increase the number of *A. muciniphila*, which directly decreases bowel inflammation. Nevertheless, the mechanism of the impact of green tea on colonocytes is unclear, although it is possible that green tea protects against colon cancer, which is a common consequence of IBD, by means of the gut microbiota modulation. This thesis was confirmed in the animal study where the administration of green tea in the course of two weeks was associated with reversing a pathological modification, including a decrease in the Firmicutes to Bacteroidetes ratio, as well as a decrease in the number of Lachinospiraceae and Ruminococcaceae, which produce SCFA, and reduce *Eubacterium* and *Roseburia* [86,87]. Moreover, the intake of green tea resulted in a decrease in the number of *Fusobacterium* in the mouth, which is beneficial for patients suffering from IBD [88]. Additionally, green tea decreased diarrhea and inhibited body mass loss, as well as decreased myeloperoxidase of the colon and TNF-α production [89].

## 6. Summary

In their guidelines, the groups investigating inflammatory bowel diseases or osteoporosis refer to the stimulants discussed in this paper:ECCO (European Crohn’s and Colitis Organization) notes that there are no clear data regarding the impact of coffee or caffeine on the risk of IBD. However, some patients, particularly individuals suffering from Crohn’s disease report avoidance of coffee, since this product exacerbates the symptoms of the disease.AACE/ACE (American Association of Clinical Endocrinologists and American College of Endocrinology) recommend limiting the consumption of beverages with caffeine to 1–2 portions per day in postmenopausal women [90].

Patients with inflammatory bowel diseases should consume beverages containing caffeine responsibly and sensibly. However, more studies are necessary regarding the impact of coffee and tea on bone metabolism, as well as concerning the course and development of osteoporosis in inflammatory bowel diseases.

## Figures and Tables

**Table 1 nutrients-13-00216-t001:** Impact of coffee compounds on the bone.

Compounds of Coffee	Activity
Caffeine	Increases urinary calcium excretion [28]
Trigonelline	Anti-estrogenic effect in an animal model [33]
Caffeic acid Chlorogenic acid Vanillic acid	Demonstrates estrogenic activity [34]

**Table 2 nutrients-13-00216-t002:** Impact of tea compounds on the bone.

Compounds of Tea	Activity
Theabrownin	Decreases osteoclastogenesis [67]
Epigallocatechin 3-gallate	Decreases the serum calcium level and urine calcium/creatinine (animal model) [68]
Theaflavins	Decreases the structure and differentiation of osteoclasts [68]

## Data Availability

Not applicable.

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
