# Peer review of "Does Drinking Coffee and Tea Affect Bone Metabolism in Patients with Inflammatory Bowel Diseases?"

_nutrients, 2021, doi:10.3390/nu13010216_

Round 1

Reviewer 1 Report

The author made many adjustments and revisions to this manuscript.

Especially there are more supplementary instructions for tea beverages.

In addition, it’s recommended to add some description in the appropriate place of this article as below:  

Eastern countries are mainly drinking green tea, semi-fermented tea and Pu-erh tea. Black tea is more popular in Western countries.

Reviewer 2 Report

All my comments in my first review were modified.
No more comment!

Author Response

This manuscript is a resubmission of an earlier submission. The following is a list of the peer review reports and author responses from that submission.

Round 1

Reviewer 1 Report

This review discussed coffee and tea affect bone metabolism in patients with inflammatory bowel diseases. According to published articles, caffeine affects BMD through many mechanisms, one of them is increase calcium urinary excretion and decreases resorption of calcium in the intestine, but daily dose of calcium are not exposed to the risk of the impact of caffeine on bone calcium metabolism. For IBD patients, the effects of caffeine on bone metabolism may be different as the inflammatory elements in intestine. However, this review does not have much references articles to discuss the bone metabolism in IBD patients. My opinion is to supplement the references in this topic; in addition, the article needs to be concise.

Comments:

On line 27: “20,2/100... “ should be 20.2 per 100,000 person-years.

Line 27: “19,2/100000” should be 19.2 per 100,000 person-years.

Line 28: “1,5 million in...” should be 1.5 million in ...,

Line 54: “2. Coffee and tea – bone metabolism” recommends using caffeine, because caffeine is mainly discussed in this section.

Line 69 to line 26: references 25 and 26 do not state what the author wants to express, it is recommended to check the references.

Line 74: “....the activity of cAPM” should be cAMP

Line 79 and 87: repeated the sentence of “caffeine increases urinary excretion of .... consumption product with caffeine” , it is recommended to consider whether it is necessary.

Line 118: “6,5%” should be 6.5%, also recommended to add a reference in line 119.

Line 125: in section 3.3, need references about IBD patients to support the relationship of coffee consumption and the risk of osteoporosis in IBD patients, same recommend in 4.3 section because in both sections do not discuss the risk of osteoporosis in IBD patients.

Line 131: “excretion by 1,6 mmol” should be 1.6 mmol/day

Line 144: need add a reference to support authors’ statement.

Line 174: “about 1,9%” should be 1.9%,

Reviewer 2 Report

Some clinical practice guidelines already recommend that people with IBD should avoid caffeine intake, though there is very little specific evidence that links caffeine to causing or worsening IBD symptoms. Some people choose to avoid it due to some of the side effects it can have which may affect their IBD.

As authors indicated that among the Asian population intake of coffee and tea was linked with a lower risk of Ulcerative Colitis (US). But the mechanism of connection is unclear.

Due to the lack of clinical evidence for the relationship between tea and caffeine intake and IBD. And this paper also lacks quantitative data or key evidence to explain the positive or negative effects of tea and coffee consumption on IBD.

However, the effect of green tea and black tea intake on intestinal health is relatively complete.